# Prevalence and determinants of erectile dysfunction among type 2 diabetes mellitus patients at selected government hospitals in gurage zone: A cross-sectional study

Seid Abrar Abdlshikure[1]*, Aklilu Getachew Mamo[2], Bisrat Fikadu[1], Abdulber Seid[3]

1 Department of Medical Laboratory Sciences, College of Medicine and Health Sciences, Wolkite University, Wolkite, Ethiopia, 2 School of Medical Laboratory Science, Faculty of Health Sciences, Institute of Health, Jimma University, Jimma, Ethiopia, 3 Department of Educational Planning & Management, College of Education & Behavioral Sciences, Wolkite University, Wolkite, Ethiopia

* seyaabfl@gmail.com

## Abstract

### Background

Erectile dysfunction (ED) is often not recognized in men with type 2 diabetes mellitus (T2DM), despite the prevalence of the disease. Early diagnosis of ED in T2DM is very important for effective treatment and prevention of serious complications such as cardio-vascular events. This study investigates the prevalence and determinants of ED among T2DM patients at selected government hospitals in the Gurage Zone.

### Methods

We conducted a study in three public hospitals from September 1 to December 30, 2020. 204 diabetic men were selected using consecutive sampling. ED was evaluated utilizing the International Index of ED-5. Glycated hemoglobin (HbA1c) and lipid levels were analyzed utilizing a Cobas 600 clinical chemistry analyzer. The data was analyzed using the Statistical Package for Social Sciences-20.

### Results

ED was observed in 156 (76.5%) diabetic males. Independent predictors of ED included age above 40 years, alcohol consumption, khat chewing, overweight [Body mass index (BMI) ≥ 25 kg/m²], obesity (BMI ≥ 30 kg/m²), a diabetes duration of more than 5 years, uncontrolled diabetes (HbA1c > 7%), elevated low-density lipoprotein (LDL), and high total cholesterol levels (P < 0.05).

### Conclusion

The study confirmed a high prevalence of ED among males with T2DM in the study settings. Consequently, clinicians should adopt preventive measures and focus on identifying the factors associated with ED in these patients.

**Data availability statement:** All relevant data are within the paper and its Supporting Information files.

**Funding:** The author(s) received no specific funding for this work.

**Competing interests:** Author have declared that no competing interest.

## Introduction

Diabetes mellitus (DM) is a global health concern, with approximately 629 million people projected to develop the disease by the year 2045, the majority being in developing countries [1]. Ethiopia ranks among the highest in Africa, with an estimated 1.3 million people living with diabetes [2].

Erectile dysfunction (ED) is defined as the inability to achieve and/or maintain an erection of sufficient rigidity and duration to permit satisfactory sexual performance [3]. Generally, ED is categorized into organic and psychogenic subtypes [4], with the organic subdivision being frequently induced by a number of causes such as diabetes mellitus (DM), hypertension, cardiovascular illnesses, and hyperlipidemia [4].

Globally, individuals with diabetes are the largest group at risk for ED, with over 422 million cases worldwide [5]. Studies indicate that diabetic men have nearly a threefold higher likelihood of developing ED compared to non-diabetic men and experience its onset 10–15 years earlier [6,7]. The prevalence of ED in men with type 2 diabetes mellitus (T2DM) is reported to be around 50%, with higher rates (e.g., 94.7% in Nigeria) documented in certain populations [8].

ED is attributed to a variety of controllable and unmodifiable causes. Age is one of the most prominent association variables for an increased incidence of various ED domains, and the condition is typically found among aged men with diabetes [9]. According to studies, the age-adjusted association factor of ED was twice as high in diabetic males as in those without diabetes [10]. Similarly, comorbid illnesses, such as hypertension, increase the risk of developing ED; an estimated 40–80% of diabetic patients with hypertension were found to have ED due to the illness itself, drug side effects, and the psychological impact of chronic diseases [11,12].

Furthermore, patients with microvascular and macrovascular diabetes problems are more likely to develop ED [13,14]. Studies have demonstrated that ED is a strong indicator of poor quality of life in patients with T2DM [15–19]. Indeed, ED has been demonstrated to be a sign of subclinical cardiovascular disease and general health, and physicians should refer these men to cardiologists, urologists, and other specialists as soon as possible to provide comprehensive care and better outcomes [9]. Despite the profound impact of ED on the lives of diabetes patients, discussing sexual health issues remains taboo in Ethiopia. As a result, sexual disorders are often overlooked and insufficiently addressed in the country [20].

Although ED is a frequently observed complication of diabetes, there is a paucity of studies on the risk of ED in T2DM in central Ethiopia [21,22]. This study was, therefore, conducted to assess the prevalence of ED among T2DM men, and to identify factors associated with ED, including their perceptions and practices regarding ED management. The findings of the study will positively influence the local decision makers to mitigate the problem through working on the identified responsible factors that contribute to ED.

## Methods and materials

### Study design, setting, and period

A hospital-based cross-sectional study was conducted from September 1 to December 30, 2020, at hospitals in the Gurage zone of central Ethiopia, 158 km south of Addis Ababa. The Gurage zone has three primary hospitals and one general hospital, all with chronic follow-up clinics for diabetes and hypertension. This study took place at the diabetic clinics of Butajira General Hospital (200 beds), Wolkite University Comprehensive Specialized Hospital (150 beds), and Attat Hospital (180 beds). These hospitals were selected due to their diverse patient populations from densely populated urban and rural areas, along with their available

resources and support. During the study, the monthly male patients diagnosed with T2DM were highest at BH (26), followed by WKUSH (15) and AH (10). A total of 104 T2DM patients were recruited from BH, 60 from WKUSH, and 40 from AH over the four months.

## Study participants

Adult male with T2DM who had a active heterosexual activity, regardless of their relationship status in the last 6 months at the time of interview and willingness to participate in the study were included in the study. Patients with ED secondary to non-diabetes-related endocrine or neurological conditions, such as hypogonadism or Parkinson's disease, and were not sexually active not because of ED for the previous 6 months prior to the study were excluded from the study (Fig 1).

## Sample size and sampling technique

The sample size for this study was determined by using a single population proportion formula. There were two previous studies conducted in Ethiopia; we have utilized the prevalence studied in the Bale zone (prevalence of 84.39%) [23], with a 5% margin of error and a standard Z score of 1.96 corresponding to 95% CI.. The sample size was computed as $n = (Z)2 p (1-p)/ d2$; $n = [(1.96)^2 * 0.8439 * (1–0.8439)]/ (0.05)^2 = 204$. The calculated sample size was 204, and, by adding a 10% non-response rate, the final sample size became 211. Among them, 3 were excluded due to non-eligibility while the remaining 4 did not give consent. Eligible male patients with T2DM attending their routine follow-up appointments at selected government hospitals were approached by investigators during the data collection period. A consecutive sampling technique was used to select all diabetic male patients who fulfill the inclusion criteria until the desired sample size is attained.

## Variables

Dependent variable. ED (Yes/No)

Independent variable.Socio-demographic factors. Age, marital status, occupational status, educational status, etc.

Behavioral and lifestyle factors. Alcohol, smoking, body mass index, khat chewing, etc.

Medical conditions. Comorbid illness, lipid profile, etc.

## Data collection

Data were collected through face-to-face interviews using a questionnaire. Participants were informed about the study's objective, and verbal and written consent was obtained.

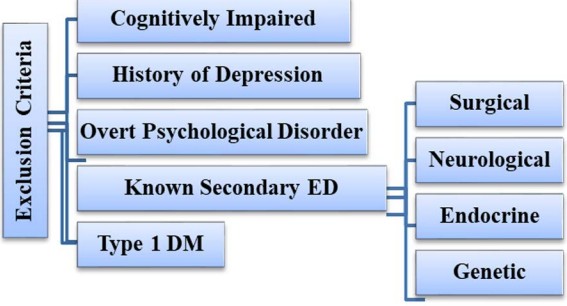

**Fig 1. A figure that shows exclusion criteria of study subjects attending selected governmental hospitals, Gurage Zone, central Ethiopia, and September 1 to December 30, 2020.**

Information collected included age, blood pressure, HbA1c, height, weight, duration of DM, hypertension, other medical conditions, smoking, alcohol use, ED treatment sought, and IIEF-5 scores. Three trained male nurses conducted the interviews on working days. The IIEF-5 questionnaire screened for ED, while biochemical measurements were taken with a COBAS 600 analyzer. A structured questionnaire assessed participants' awareness and attitudes toward ED treatments, including perceived medication effectiveness and sexual counseling. Data collection took place in a separate, quiet room. Patients identified with ED were informed of their condition and advised to consult their physician.

## Operational definition

Glycemic control: The level of glycemic control was assessed by obtaining HbA1C values. Participants were grouped into normal glycemic control (i.e., HbA1C ≤ 7) and poor control (i.e., HbA1C > 7) [24].

Alcoholic: The daily alcohol amount that respondents consume was calculated considering the average alcohol percent (%/ml) of each drink multiplied by the volume (ml) of the drink and volumetric mass density (which is 0.8 g/ml). Accordingly, participants were explained to be alcoholic if they drank more than 12g of ethanol of alcohol per day in the past six months of the survey [25].

Cigarette smoking: those who smoke any tobacco products daily will be considered tobacco users [26].

Physical inactivity: Those respondents who will not achieve WHO recommendations of total physical activity (level less than 600 METs-minute/week) or reported practicing no physical activities will be classified as inactive. Whereas those who meet this criterion (> 600 METs-minutes/week) will be classified as physically active [27]

Khat chewer: if the patient reported khat chewing (kata adulis forsik)

Comorbidity: any medical condition concomitantly occurring with diabetes mellitus. Example: hypertension [28].

Diabetic complications: the existence of macrovascular diabetic complications, microvascular diabetic complications (retinopathy, neuropathy, and nephropathy), and diabetic foot ulcer.

Body mass index (BMI): a person's weight in kilograms (kg) divided by his or her height in meters squared (m2). Based on WHO's 2004 BMI classification, participants who have a BMI (kg/m$^2$) < 18.5, 18.5–24.9, 25–29.9, and > 30 kg/m$^2$ will be classified as underweight, normal range, overweight, and obese, respectively [29].

ED is the outcome variable for this study. It was assessed using the International Index of Erectile Function (IIEF) questionnaire. This questionnaire consists of five questions, each with Likert scale alternative responses. The greatest possible score is 25, which is the score for people who never had an erection problem, and the lowest score is 5. The sum of the ordinal replies to the 5 items determines the IIEF-5 score. The ED was categorized into two depending on the IIEF score. Those who score 22 and above were classified as having no ED, and those who score 21 and below were classified as having ED. More specifically, those who have a score of 17–21, 12–16, 8–11, and 5–7 are subcategorized as having mild, mild to moderate, moderate, and severe ED, respectively [30].

## Data processing, analysis and quality assurance

The questionnaire and checklist were reviewed by experts and translated into Amharic, followed by back-translation for consistency. To ensure data quality, facilitators and data collectors were trained, and the tool was pretested on 10% of participants. Data collection occurred in a private room, with daily supervision of data collectors by investigators.

Data were collected using the Amharic version, checked for consistency, errors, completeness, accuracy, and clarity before being entered into Epi-data version 3.1. The data were then exported to SPSS version 20 for recoding, cleaning, and analysis. All continuous independent variables were categorized.

The outcome variable was dichotomized and coded as '0' and '1', representing those who have no and have ED, respectively. Continuous data were checked for normality with the Shapiro–Wilk test and were found not to be normally distributed. Descriptive statistics like frequency, percentage, and measure of central tendency with their corresponding measure of dispersion were used to describe demographic and other variables. Tables', graphs, and texts were used to present the findings.

Furthermore, the binary logistic regression analysis was applied to identify factors associated with ED. Those variables with a p-value ≤0.25 in the bi-variable analysis were entered into the multi-variable logistic regression model to control the possible effects of confounder/s and to identify the significant factors. According to the Hosmer and Lemeshow test, the model was found to be adequate. Likewise, prior to identifying the significant factors, the presence of mul-ticollinearity problem was examined using the Variance Inflation Factor (VIF), and no variable was found to have that problem.

## Ethical consideration

Ethical clearance to conduct the study was obtained from the institutional health research ethical review committee of Jimma University. This ethical clearance was taken to the AH, BH, and WKUCSH clinical director office, and a permission letter to conduct the study was taken from the clinical director office to the AH, BH, and WKUCSH diabetic clinic to conduct the study. The objective of the study was explained to each study participant. Verbal and written informed consent was obtained from each respondent before the interview. The consent was taken after participants were informed about the risk, benefit, and their right to withdraw from the study at any time during the interview process. Those volunteers who signed on the consent form were involved in the study. Moreover, all information taken from the respondents was kept confidential, and the entire data collected was only used for the purpose of this study.

## Results

### Socio-demographic and clinical characteristics of participants

A total of 204 participants were enrolled study making a response rate of 96.68%. The mean (±standard deviation (SD)) age of the respondent was 44.09(±13.04) years. The majority of participants, specifically 73.53%, were married. Almost more than half of them, i.e., 52.94%, were from families with a monthly income of more than 2000 Ethiopian Birr (equivalent to 50.8 USD). The study also revealed that 32.84% of participants were smokers, 43.6% were alcohol consumers, and more than half (52.94%) were khat chewers (Table 1).

Table 2 shows, 43.1% of the study participants had diabetes for 5–10 years, and 68.6% of them used tablets for treatment. Most of the participants (65.2%) had uncontrolled diabetes, that is, their A1c levels were higher than 7. About 34.8% of them were found to be obese. The total cholesterol level (>200 mg/dl) was abnormal in more than half of the patients.

### Prevalence of ED among the study participants

As Fig 2 shows, the prevalence rate of ED among T2DM was 76.5% (156 out of 204).

**Table 1. Socio-demographic characteristics among T2DM attending selected governmental hospitals, Gurage zone, central Ethiopia, and September 1 to December 30, 2020. (n = 204).**

| Socio-Demographic Variables | Category | Frequency(%) |
|---|---|---|
| Age Group | <40 Years | 73(35.79%) |
| | 40–50 Years | 72(35.29%) |
| | >50 Years | 59(28.92%) |
| Marital Status | Single | 41(20.10%) |
| | Married | 150(73.53%) |
| | Separated | 12(5.88%) |
| | Divorced | 1(0.49%) |
| Educational Status | No Formal Education | 48(23.53%) |
| | Primary | 47(23.04%) |
| | Secondary | 37(18.14%) |
| | Diploma | 34(16.67%) |
| | Degree And Above | 38(18.62%) |
| Occupational Status | Farmer | 50(24.51%) |
| | Merchant | 57(27.94%) |
| | Government Employee | 58(28.43%) |
| | Non Employed | 39(19.12%) |
| Monthly Income (Ethiopian Birr) | <1000 | 35(17.16%) |
| | 1000–2000 | 61(29.90%) |
| | >2000 | 108(52.94%) |
| Physical Activity | Yes | 83(40.69%) |
| | No | 121(59.31%) |
| Alcohol Drink | Yes | 89(43.63%) |
| | No | 115(56.37%) |
| Smoking | Yes | 67(32.84%) |
| | No | 137(67.16%) |
| Khat Chewing | Yes | 108(52.94%) |
| | No | 96(47.06%) |

## ED acceptance and treatment-seeking rate

The results presented in **Table 3** align with the study's aim by providing insights into participants' perceptions of ED treatments, which are essential for understanding the barriers to seeking care and improving management practices.

Most of the patients who were included in the study (94.12%) were not screened or managed for ED. A significant number, 23 (11.27%), of study participants think that drugs are effective in treating ED cases. In the same manner, only 37 (18.14%) of the study participants think that sexual counseling is effective in the treatment and prevention of ED (**Table 3**).

## Factors associated with ED

In multi-variable analysis variables such as older age, long duration of diagnosed with DM, alcohol drink,khat chewing,high BMI, poor glycemic control, and dyslipedemia (elevated LDL), and high TC levels) have shown an independent association with ED (**Table 4**).

The association between ED and participant demographics along with clinical characteristics. ED was associated with 73.7% of participants 40 and above years (p < 0.001), alcohol consumption (p < 0.001), and chat consumption (p < 0.001). Participants with a higher

**Table 2. Participants' clinical characteristics among T2DM men attending selected governmental hospitals, september 1 to december 30, 2020. (n = 204).**

| Clinical variables | Category | Frequency(%) |
|---|---|---|
| BMI | Normal | 49(24.02%) |
| | Over weight | 84(41.18%) |
| | obese | 71(34.80%) |
| Duration of diabetes | <5yrs | 79(38.73%) |
| | 5–10yrs | 88(43.13%) |
| | >10yrs | 37(18.14%) |
| Diabetes complication | Yes | 58(28.43%) |
| | No | 146(72.57%) |
| Co-existing hypertension | Yes | 92(45.10%) |
| | No | 112(54.90%) |
| On hypertensive drug | Yes | 67(32.84%) |
| | No | 137(67.16%) |
| Glycemic control | Good | 71(34.80%) |
| | Poor | 133(65.20%) |
| HDL | Normal(≥40 Mg/Dl) | 104(50.98%) |
| | Abnormal(<40mg/Dl) | 100(49.02%) |
| LDL | Normal(≤100mg/Dl) | 113(55.39%) |
| | Abnormal(>100mg/Dl) | 91(44.61%) |
| TC | Normal (≤200mg/Dl) | 84(41.18%) |
| | Abnormal (>200mg/Dl) | 120(58.82%) |
| TG | Normal (≤150mg/Dl) | 92(45.10%) |
| | Abnormal (>150mg/Dl | 112(54.90%) |

NOTE: BMI: Body Mass Index, DM: Diabetes Mellitus, HDL (High Density Lipoprotein), LDL (Low Density Lipo-protein), TC (Total Cholesterol), TG (Triglyceride)

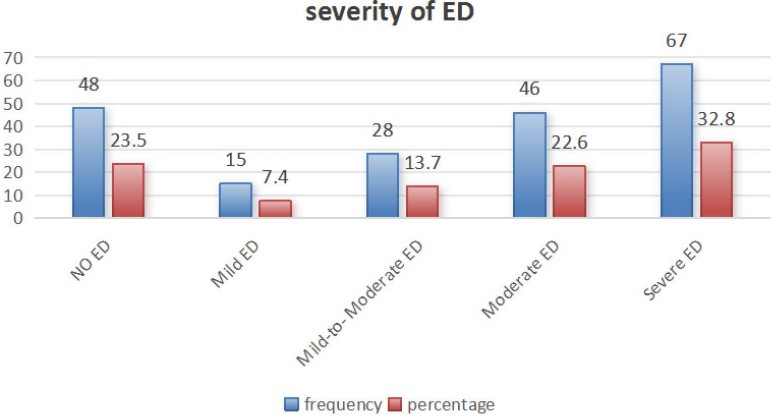

**Fig 2. Erectile function scores of T2DM men attending selected governmental hospitals, Gurage zone, central Ethiopia, and September 1 to December 30, 2020. (n = 204).**

BMI (≥25 kg/m²) were more likely to experience ED (p < 0.001). In addition, abnormal HDL, abnormal LDL, and high TC levels were associated with ED (p < 0.001). Furthermore, 91.6% of participants who had diabetes for 5 years or more experienced ED (p = 0.002). As

**Table 3. Acceptance and treatment-seeking rate for ED among T2DM men attending selected governmental hospitals, Gurage zone, central Ethiopia, and September 1 to December 30, 2020. (n = 204).**

| Acceptance and treatment-seeking rate for ED | ED | No ED | Total Frequency(%) |
|---|---|---|---|
| History of seeking medical help | | | |
| Yes | 12(100.0%) | 0 | 12(5.88%) |
| No | 144(75.0%) | 48(25.0%) | 192(94.12%) |
| Treated for sexual problem | | | |
| no | 145(75.1%) | 48(24.9%) | 193(94.61%) |
| yes | 11(100.0%) | 0 | 11(5.39%) |
| Thinking of drug are effective | | | |
| Yes | 14(60.9%) | 9(39.1%) | 23(11.27%) |
| No | 142(78.5%) | 39(21.5%) | 181(88.73%) |
| Thinking of sexual counseling is effective | | | |
| Yes | 17(45.9%) | 20(54.1%) | 37(18.14%) |
| No | 139(83.2%) | 28(16.8%) | 167(81.86%) |

mentioned above, there was no significant association between ED and marital status, occupation, monthly income, physical activity, smoking status, hypertensive drug use, educational status, abnormal triglyceride level, coexisting hypertension, and diabetes complications(S Table 1).

## Discussions

ED has been reported to be an important cause of decreased quality of life in men [3]. As socio-cultural changes can veil the prevalence and determinants of ED, in this regard, many studies have investigated the prevalence and determinants of ED as the interest in the disease increased. However, little has been done in Sub-Saharan Africa and Ethiopia on this issue. To the best of our knowledge, information concerning contributing factors of ED among T2DM patients was uncommon.

The prevalence of ED among patients with T2DM was found to be 76.5% in the present study. This is in line with studies conducted in different parts of the world ranging between 35 and 90%. Moreover, the study finding coincides with the pooled results from India, 68.7% [31] and Africa, 71.45% [32]. However, there is some discrepancy with the study conducted in Ethiopia, 54.3% [33]. The reason for these discrepancies in prevalence rates across studies is not scientifically explained, though some authors do provide possibilities. Factors such as population sizes, demographic characteristics, duration and severity of diabetes, and the presence of other comorbidities may account for these divergent rates.

A consistent risk factor for ED includes increasing age, in both the diabetic and general populations [34]. However, in men with diabetes, ED is more frequent than in men of the same age who do not have diabetes [35]. The onset of ED also occurs 10–15 years earlier in men with diabetes than it does in those without diabetes [36].n our study, the odds of having ED were 5.8 and 6.9 times higher among participants who were aged 40–50 and >50 years than in younger participants, respectively. The advancing age of diabetes has been consistently documented to increase the probability of having ED, which is a finding that is confirmed by the work of Tridiantari et al. and Ghanem et al. [18,37].

There are some controversial results in another study where increasing age is not an independent determinant of ED in diabetes [38]. Increasing age poses the chance of having ED due to physiological changes of atherosclerosis and the consequently reduced blood flow to the sex organs [39]. Although the aging process is inevitable, preventive measures such as

**Table 4. Independent predictors of ED among T2DM men attending selected governmental hospitals, Gurage zone, central Ethiopia, and September 1 to December 30, 2020. (n = 204).**

| Variable | Category | ED status | | AOR (Lower-Upper) | P-value |
|---|---|---|---|---|---|
| | | NO | YES | | |
| Age group | < 40 Years | 32(66.67%) | 32(20.51%) | 1 | 1 |
| | 40–50 Years | 14(29.17%) | 46(29.49%) | 5.80 (2.08–16.76) | 0.000* |
| | >50 Years | 2(4.16%) | 110(70.51%) | 6.90 (1.79–11.98) | 0.000* |
| Marital Status | Single | 5(31.25%) | 26(16.67%) | 1 | 1 |
| | Married | 30(62.50%) | 120(76.92%) | 0.60 (0.14–2.52) | 0.482 |
| | Separated | 3(6.25%) | 10(6.41%) | 1.01 (0.12–8.52) | 0.991 |
| Occupational Status | Farmer | 10(20.83%) | 40(25.64%) | 1 | 1 |
| | Merchant | 12(25.00%) | 45(28.85%) | 0.74 (0.15–3.83) | 0.724 |
| | Government | 12(25.00%) | 46(29.49%) | 0.93 (0.21–4.06) | 0.918 |
| | Non Employed | 14(29.17%) | 25(16.03%) | 0.51 (0.09–2.87) | 0.442 |
| Monthly Income (Birr) | <1000 | 12(25.00%) | 23(14.74%) | 1 | 1 |
| | 1000–2000 | 16(33.33%) | 45(28.85%) | 1.41(.270–7.368) | 0.683 |
| | >2000 | 20(41.67%) | 88(56.41%) | 1.75 (0.29–10.72) | 0.542 |
| Alcohol drink | Yes | 10(20.83%) | 79(50.64%) | 12.68 (2.96–54.40) | 0.001* |
| | No | 38(79.17%) | 77(49.36%) | 1 | 1 |
| Smoking | Yes | 11(22.92%) | 56(35.90%) | 1.51 (0.47–4.84) | .492 |
| | No | 37(77.08%) | 100(64.10%) | 1 | 1 |
| Khat Chewing | Yes | 15(31.25%) | 93(59.62%) | 5.29 (1.62–17.28) | 0.006* |
| | No | 33(68.75%) | 63(40.38%) | 1 | 1 |
| BMI | Normal | 25(52.08%) | 24(15.38%) | 1 | 1 |
| | Over Weight | 10(20.83%) | 74(47.44%) | 8.49 (2.33–30.88) | 0.001* |
| | Obese | 13(27.08%) | 58(37.18%) | 5.85 (1.54–22.27) | 0.010* |
| Duration of diabetes | <5yrs | 29(60.41%) | 50(32.05%) | 1 | 1 |
| | ≥5yrs | 19(39.59%) | 106(67.95%) | 3.28 (1.06–10.19) | 0.039* |
| Glycemic control | Good | 29(60.42%) | 42(26.92%) | 1 | 1 |
| | Poor | 19(39.58%) | 114(73.08%) | 4.23 (1.36–13.14) | 0.013* |
| HDL | Normal(≥40 Mg/Dl) | 36(75.00%) | 68(43.59%) | 1 | 1 |
| | Abnormal(<40mg/Dl) | 12(25.00%) | 88(56.41%) | 2.22 (0.71–6.96) | 0.171 |
| LDL | Normal(≤100mg/Dl) | 40(83.33%) | 73(46.79%) | 1 | 1 |
| | Abnormal(>100mg/Dl) | 8 (16.67%) | 83(53.21%) | 4.59 (1.32–15.92) | 0.016* |
| TC | Normal (≤200mg/Dl) | 35(72.92%) | 49(31.41%) | 1 | 1 |
| | Abnormal(>200mg/Dl) | 13 (27.08%) | 107(68.59%) | 4.16 (1.35–12.81) | 0.013* |

NOTE: AOR: Adjusted Odds Ratio BMI: Body Mass Index, CI: Confidence Interval DM: Diabetes Mellitus, HDL: High-Density Lipoprotein, LDL: Low-Density Lipoprotein, TC: Total Cholesterol 1: Reference category,

*: significant p < 0.005, Hosmer and Lemshow goodness of fit (p-value = 0.44), Multicollinearity test (VIF) = 1.36

adopting a healthy lifestyle and eating habits can reduce the future burden of this problem [40]. Seeking doctors' help and early treatment measures may also help to mitigate the onset of ED.

A meta-analysis of population-based cross-sectional studies has found significantly decreased odds of ED among alcohol consumers [41]. In the present study, the odds of having ED among alcohol consumers were 12.6 times higher than among non-alcohol consumers. This result is similar to the study conducted in Gabon and India [42,43]. One possible

explanation for the increase in ED following alcohol consumption could be that alcohol triggers the RhoA/Rho-kinase pathway, causes increased oxidative stress, and increases the penis' reactivity to the constrictor effects of endothelin-1. Endothelin-1 (ET-1) is a potent vasoconstrictor in the penis, and its vasoconstriction is linked to the RhoA/Rho-kinase pathway, which reduces the production of nitric oxide (NO) by suppressing endothelial nitric oxide synthase (eNOS) [44].

Although the effect of smoking has also been shown as the cause for ED in T2DM [45], this was not observed in our study. Our findings are in line with other reports where smoking habit was not related to the prevalence of ED [46,47]. The lower number of respondents who identified as smokers could explain why the findings do not align with those of previous studies.

Furthermore, the probability of having ED among Khat chewers was 5.3 times higher than non-Khat chewers. The possible reason for the high chance of ED among Khat chewers compared to non-Khat chewers may be due to Khat chewing affecting male sexual potency by inhibiting spermatogenesis and reducing testosterone levels [48]. However, the work of Asefa et al. and Nassar et al. failed to show chewing khat as an independent predictor for ED in diabetes [49,50]. Further research is needed to determine whether chewing khat contributes to the morbidity of ED.

Previous research findings indicate that diabetic patients tend to have significantly higher BMI and waist circumference compared to non-diabetic individuals [51]. It is well-known that obesity is a risk factor for ED in diabetic patients [52].In our study, we found that excess weight is strongly associated with a higher likelihood of developing ED, showing an 8.5-fold increase. Similarly, obesity is linked to a 5.8-fold increase in the chance of experiencing ED. Studies in the past have shown similar findings and have shown a direct relationship between obesity and ED [8,16]. Subclinical inflammation and oxidative stress in obesity lead to leptin resistance that results in low LH and low testosterone, which is one of the causes of ED [53]. Insulin resistance in obesity elevates oxidative stress and inflammatory cytokines, such as TNF-α and IL-6, in endothelial cells, which decreases nitric oxide bioavailability and induces endothelial dysfunction [54].

The duration of diabetes is the most significant factor associated with the occurrence of ED in T2DM patients. If the duration of diabetes is greater than 5 years, the risk of association with ED triples. This is in line with the findings of previous studies conducted in Sri Lanka and Gabon [43,55].

In our study, dyslipidemia characterized by showed higher levels of total cholesterol and abnormal lipoprotein levels. The probability of having ED was 5.2 and 5.6 times higher in participants with high serum TC and LDL levels than in those who had normal serum TC and LDL levels, respectively. This finding is in line with a previous study conducted by Fedele et al. in Italy [56]. This result was also noted by Azad et al.; it was observed that there is a correlation between abnormal LDL, high TC, and ED [57].

In addition, the probability of having ED was 5.2 times higher in men with poor blood sugar compared to those with well-controlled blood sugar. This finding highlights the critical role of maintaining optimal blood glucose levels in reducing the chance of ED and improving overall health outcomes.This finding is consistent with studies conducted in India and Saudi Arabia [42,58].

The study also presented no significant relationship between any of the sexual domains with hypertension, hyperlipidemia, and high TG in men with T2DM. These findings were consistent with Sharifi's study, which found that men with T2DM had no significant association with blood pressure and levels of triglycerides [59].

When considering the medications, no significant association between being on antihypertensive drugs and ED was noted. This finding is congruent with a study conducted in Nigeria [60]. However, this data was contradicted with the study in Zimbabwe [61]. Also, on the contrary to our outcome, recently published articles in Ethiopia revealed that diabetic

complications (AOR=2.05,95% CI: 1.18–3.58, P=0.01) were significantly associated with sexual dysfunction among diabetes patients [49].

In this research, it was noteworthy that a significant percentage (94.12%) of participants had not addressed the issue or pursued medical treatment for ED from a physician, including those dealing with DM. This concerning pattern has been reported in Asia [62], where a majority of patients had neither sought nor received treatment for ED (fewer than 10% had undergone treatment). The low rate of seeking treatment may be attributed to patient-related obstacles, such as feelings of shame or embarrassment, culturally unsuitable services, and a lack of awareness [63].

This study didn't find a link between smoking, income, education, occupation, physical activity, diabetes complications, hypertension medication, high triglyceride levels, low HDL, and ED, possibly due to cultural factors. The study highlights the high prevalence of ED among T2DM patients and identifies several determinants. It calls for awareness campaigns for patients, the public, and the government.

## Strength and limitation of the study

This study utilized standard tools, such as IIEF, but may have experienced over-reporting of cases due to participants hoping for better management. Despite efforts to minimize bias, including male interviewers and private settings, social desirability bias may still have influenced responses due to the sensitive nature of the topics discussed. Additionally, the inability to measure participants' testosterone levels limits the analysis of diabetes' impact on men's sexuality. We also lacked a control group to compare the prevalence of ED between the general population and T2DM patients. The cross-sectional design precludes causal inferences about the risk factors for ED in the studied population. Future research should consider using continuous variable values to enhance the robustness of the analysis.

## Conclusion

In conclusion, the prevalence of erectile dysfunction among adult T2DM is nearly eight out of ten. ED health-seeking behavior was poor, with only a minority of patients seeking and receiving care. To manage men with T2DM, we recommend objective ED screening with standard brief instruments, early glycemic goal achievement through lifestyle interventions, culturally appropriate ED-related conversations in private, and referral to an ED clinic for urological input. The Ethiopian Ministry of Health should incorporate ED management into diabetes guidelines. Future studies on ED among diabetics should consider using self-administered questionnaires or digital survey tools to gather data.

## Supporting information

**S1 Table. Association between variables among T2DM.**
(DOCX)

**S1 File. Questionnaire.**
(DOCX)

## Acknowledgments

Special thanks to the BH, WKUSH, and AH staffs for their intensive cooperation in sample analysis. Finally, we would like to appreciate to all the study participants to provide their valuable time and data for the research.

## Author contributions

**Conceptualization:** Seid Abdlshikure.

**Data curation:** Bisrat Fikadu.

**Formal analysis:** Abdulber Seid.

**Investigation:** Bisrat Fikadu.

**Supervision:** Aklilu Getachew Mamo.

**Visualization:** Seid Abdlshikure.

**Writing – original draft:** Seid Abdlshikure.

**Writing – review & editing:** Seid Abdlshikure.

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
