## [Decision Letter · Decision Letter 0]

1 Sep 2024

PONE-D-24-27116Prevalence and Determinants of Erectile Dysfunction among Type 2 Diabetes Mellitus patients at Selected Government Hospitals in Gurage ZonePLOS ONE

Dear Dr. ABRAR,

Thank you for submitting your manuscript to PLOS ONE. After careful consideration, we feel that it has merit but does not fully meet PLOS ONE’s publication criteria as it currently stands. Therefore, we invite you to submit a revised version of the manuscript that addresses the points raised during the review process.

ACADEMIC EDITOR: Thanks for submitting this manuscript to this esteemed journal. Based on the 3 reviews especially addressing the novelty about what this research adds to what is already known i suggest drafting and major revision Thanks ;

We look forward to receiving your revised manuscript.

Kind regards,

Kamal Sharma

Academic Editor

PLOS ONE

Journal Requirements: When submitting your revision, we need you to address these additional requirements. 1. Please ensure that your manuscript meets PLOS ONE's style requirements, including those for file naming. The PLOS ONE style templates can be found at https://journals.plos.org/plosone/s/file?id=wjVg/PLOSOne_formatting_sample_main_body.pdf and https://journals.plos.org/plosone/s/file?id=ba62/PLOSOne_formatting_sample_title_authors_affiliations.pdf 2. We note that the grant information you provided in the ‘Funding Information’ and ‘Financial Disclosure’ sections do not match.  When you resubmit, please ensure that you provide the correct grant numbers for the awards you received for your study in the ‘Funding Information’ section. 3. Thank you for stating the following financial disclosure: "This research work was supported by Jimma university school of medical laboratory sciences, faculty of health science, and institute of health. The research received no funding and no specified grant number." Please state what role the funders took in the study.  If the funders had no role, please state: ""The funders had no role in study design, data collection and analysis, decision to publish, or preparation of the manuscript."" If this statement is not correct you must amend it as needed. Please include this amended Role of Funder statement in your cover letter; we will change the online submission form on your behalf. 4. Your ethics statement should only appear in the Methods section of your manuscript. If your ethics statement is written in any section besides the Methods, please move it to the Methods section and delete it from any other section. Please ensure that your ethics statement is included in your manuscript, as the ethics statement entered into the online submission form will not be published alongside your manuscript. 5. Please ensure that you refer to Figure 1 in your text as, if accepted, production will need this reference to link the reader to the figure. 6. Please include a copy of Table 3 and 5 which you refer to in your text on page 13 and 16. 7. Please include captions for your Supporting Information files at the end of your manuscript, and update any in-text citations to match accordingly. Please see our Supporting Information guidelines for more information: http://journals.plos.org/plosone/s/supporting-information.

**Additional Editor Comments:**

Hello,

Based on the reviewers comments the manuscript needs to undergo a major revision addressing all the issues raised by all the 3 reviewers

Thanks

Reviewers' comments:

Reviewer's Responses to Questions

Comments to the Author

1. Is the manuscript technically sound, and do the data support the conclusions?

Reviewer #1: Yes

Reviewer #2: Yes

Reviewer #3: Yes

2. Has the statistical analysis been performed appropriately and rigorously?

Reviewer #1: Yes

Reviewer #2: No

Reviewer #3: Yes

3. Have the authors made all data underlying the findings in their manuscript fully available?

Reviewer #1: Yes

Reviewer #2: Yes

Reviewer #3: Yes

4. Is the manuscript presented in an intelligible fashion and written in standard English?

Reviewer #1: Yes

Reviewer #2: Yes

Reviewer #3: Yes

5. Review Comments to the Author

Reviewer #1: Please, in the abstract, method, instead of 204 diabetic patients, the wording should be corrected as 204 diabetic men. In the results, the same should be corrected.

Two age groups under and over 40 years were considered. Please divide the age group above 40 years into subgroups and compare the prevalence between them.

In the results, instead of blood sugar above 7, write A1C. Because this is how the method is perceived.

Limitations should be mentioned.

Reviewer #2: General comments

This is an interesting study which highlighted the prevalence and predictors of ED among male patients with diabetes in three hospital in Ethiopia. However, a few issues may need to be addressed to strengthen the paper.

Specific comments

1.Page 6, sample size determination: How representative is a convenience sample of 204 male type 2 DM patients in the study areas? It will be important to state the estimated number of male type 2 DM patients that usually attend the AH, BH and WKUSH hospitals in a four months period. Please include the sample size formula and the variables used for the calculation of sample size, so the reader can follow your calculation.

2.Page 7, exclusion criteria, figure 1:

a.1st line: The statement “Study participants with the following disorder will exclude from the study” should read “Study participants with the following disorders were excluded from the study”

b.“Exclusion criterias” should read “Exclusion criteria” since criteria is already a plural word.

c.Menatlly Incompitent: has spelling error. Should read “Mentally incompetent”. However, what do you mean by mentally incompetent? Please rephrase for clarity.

d.History of Deperession: should read “History of Depression”.

e.Overt Psychlogical Disorder: should read “Overt Psychological Disorder”

f.“Nurological” should be replaced by “Neurological”

3.Page 9. Data collection, line 3: what type of consent did you obtain from the participants? Was it written or verbal? This should be clearly stated.

4.Page 9, 4.7.1.1, socio-demographic, anthropometric and related clinical data: It will be important to clearly define smoking status, alcohol consumption, exercise status, and income. What is currently stated is still unclear. For instance, when you say smoker and non-smoker, what do you mean? Do you mean never ever smoked, stopped smoking 6 months ago, actively smoking, etc. There are standard definitions for these words and needs to be stated as such and also cite appropriate references from the literature to back your definitions. This will help the reproducibility of the study.

RESULTS

5.Table 1, educational and occupational statuses, frequency columns do not total 100%. Please cross check.

6.Page 13, 5.3 prevalence of erectile dysfunction among the study participants:

a.line 1: If 204 male T2DM patients were interviewed, how come 216 had T2DM and ED? Do you mean 156 instead of 216? Please cross-check.

b.Lines 3-6: “Further analysis of the data showed that 48 men had no ED, 15 had mild ED, 28 had mild-to-moderate ED, 46 had moderate ED, and 67 had severe ED. The data also indicated that a maximum of 32.8% of men with diabetes had severe ED, while only 7.4% had a mild form of ED..” It will be important to include both the numbers and the full percentages of the ED severity types because it appears the figure presented shows summarized percentages.

7.Page 14, Table 4: what test was employed to determine the association between clinical variables and ED among study participants? From this table, it appears you applied a different statistical test other than bivariate logistic regression. It seems you used a chi-square test. Please clarify and possibly re-write your section 5.4.

DISCUSSION

8.Page 19, paragraph 3: No need to restate the AOR, 95%CI and P value in the discussion. This also applies to the last paragraph and elsewhere.

9.Strength and limitations of the study: the convenience sampling technique presents a limitation towards generalising the study result. Given the selection bias, those who participated may be significantly different from those who did not.

Reviewer #3: This is a work about the prevalence and determinants of ED among T2D patients at Selected Government Hospitals in Gurage Zone. Anyway, no novelties about this topic are producted. I have comments and suggestions.

Abstract

- please, as suggested by international guidelines, replace here and in the manuscript, diabetic with subjects/patients with diabetes (T2D).

Introduction

- "it remains uncertain whether erectile dysfunction (ED) in diabetic males stems

solely from hyperglycemia and microvascular complications or arises from a combination of

factors such as cardiovascular diseases, hypertension, smoking, and obesity". Vs 2010, now we have more scientific data to prove these combinations.

Materials and methods ( very long! )

Inclusion/exclusion criteria: You insert T2D as an exclusioon criteria. Please, check!

Discussion

- some parts of results are here too.

- please, clarify better strenghts and limitations of this work, not after conclusions section.

Editorial guidelines needs to be checked.

Check acronyms as type 2 diabetes

6. PLOS authors have the option to publish the peer review history of their article (what does this mean? ). If published, this will include your full peer review and any attached files.

Do you want your identity to be public for this peer review? For information about this choice, including consent withdrawal, please see our Privacy Policy .

Reviewer #1: Yes: Maryam Zahedi, Assistant Professor of Endocrinology & Metabolism,

Clinical Research Development Unit (CRDU),

Sayad Shirazi Hospital,

Golestan University of Medical Sciences,

Gorgan, Iran.

Tel: +98 9111778058

Email: drmaryam.zahedi@yahoo.com

Reviewer #2: Yes: Godpower Chinedu Michael

Reviewer #3: No

---

## [Author Response · Author response to Decision Letter 1]

22 Sep 2024

Reviewer #1 Comments and response

Reviewer #1: Please, in the abstract, method, instead of 204 diabetic patients, the wording should be corrected as 204 diabetic men. In the results, the same should be corrected.

Response to reviewer #1: Thank you, dear reviewer. We have now revised the content as per suggestion.

Two age groups under and over 40 years were considered. Please divide the age group above 40 years into subgroups and compare the prevalence between them

response to reviewer #1: We have now revised the content as per suggestion, the age grouped .< 40 years ,40-50 years and >50 years and the prevalence were computed

In the results, instead of blood sugar above 7, write A1C. Because this is how the method is perceived.

Limitations should be mentioned.

response to reviewer #1: We have now revised the content as per suggestion and the Limitations also were mentioned.

Reviewer #2 Comments and response

This is an interesting study which highlighted the prevalence and predictors of ED among male patients with diabetes in three hospital in Ethiopia. However, a few issues may need to be addressed to strengthen the paper.

Specific comments

1.Page 6, sample size determination: How representative is a convenience sample of 204 male type 2 DM patients in the study areas? It will be important to state the estimated number of male type 2 DM patients that usually attend the AH, BH and WKUSH hospitals in a four months period. Please include the sample size formula and the variables used for the calculation of sample size, so the reader can follow your calculation.

Response to reviewer #2: We have now revised the content as per suggestion. estimated number of male type 2 DM patients Over the 4-month study period, 104 type 2 diabetic patients were recruited from BH, 60 from WKUSH, and 40 from AH. the sample size was computed as:

n= (Z) 2 p (1-p) /d2 ; n= [(1.96) ²* 0.8439*(1-0.8439)]/(0.05)2 = 204

2.Page 7, exclusion criteria, figure 1:

a.1st line: The statement “Study participants with the following disorder will exclude from the study” should read “Study participants with the following disorders were excluded from the study”

Response to reviewer #2:Thank you, we have now corrected

b.“Exclusion criterias” should read “Exclusion criteria” since criteria is already a plural word.

Response to reviewer #2:Thank you, we have now corrected

c.Menatlly Incompitent: has spelling error. Should read “Mentally incompetent”. However, what do you mean by mentally incompetent? Please rephrase for clarity.

Response to reviewer #2: Thank you, we have now corrected and repharesed to “Cognitively impaired”

d.History of Deperession: should read “History of Depression”.

Response to reviewer #2: Thank you, we have now corrected

e.Overt Psychlogical Disorder: should read “Overt Psychological Disorder”

Response to reviewer #2: Thank you, we have now corrected

f.“Nurological” should be replaced by “Neurological”

Response to reviewer #2: Thank you, we have now corrected .

2.Page 9. Data collection, line 3: what type of consent did you obtain from the participants? Was it written or verbal? This should be clearly stated.

Response to reviewer #2: We have now revised the content as per suggestion oral consent were obtained from the participants and clearly stated

3.Page 9, 4.7.1.1, socio-demographic, anthropometric and related clinical data: It will be important to clearly define smoking status, alcohol consumption, exercise status, and income. What is currently stated is still unclear. For instance, when you say smoker and non-smoker, what do you mean? Do you mean never ever smoked, stopped smoking 6 months ago, actively smoking, etc. There are standard definitions for these words and needs to be stated as such and also cite appropriate references from the literature to back your definitions. This will help the reproducibility of the study.

Response to reviewer #2: Thank you, we have now included standard definitions of the words and stated as such and cited with appropriate references

RESULTS

4.Table 1, educational and occupational statuses, frequency columns do not total 100%. Please cross check.

Response to reviewer #2: Thank you, We have now cross checked and corrected

6.Page 13, 5.3 prevalence of erectile dysfunction among the study participants:

a.line 1: If 204 male T2DM patients were interviewed, how come 216 had T2DM and ED? Do you mean 156 instead of 216? Please cross-check.

Response to reviewer #2: Thank you, We have now revised and corrected to 156

b.Lines 3-6: “Further analysis of the data showed that 48 men had no ED, 15 had mild ED, 28 had mild-to-moderate ED, 46 had moderate ED, and 67 had severe ED. The data also indicated that a maximum of 32.8% of men with diabetes had severe ED, while only 7.4% had a mild form of ED..” It will be important to include both the numbers and the full percentages of the ED severity types because it appears the figure presented shows summarized percentages.

Response to reviewer #2: Thank you, We have now revised and included both the numbers and the full percentages of the ED severity

The prevalence rate of ED among Type 2 diabetic men was 76.5% (156 out of 204). Further analysis of the data showed that 48(23%) men had no ED, 15(7%) had mild ED, 28(14%) had mild-to-moderate ED, 46(23%) had moderate ED, and 67(33%) had severe ED

7.Page 14, Table 4: what test was employed to determine the association between clinical variables and ED among study participants? From this table, it appears you applied a different statistical test other than bivariate logistic regression. It seems you used a chi-square test. Please clarify and possibly re-write your section 5.4.

Response to reviewer #2: Thank you, We have now revised and corrected

Chi-square test was carried out to discover the relationship between two categorical variables and re-wrote in the section

DISCUSSION

8.Page 19, paragraph 3: No need to restate the AOR, 95%CI and P value in the discussion. This also applies to the last paragraph and elsewhere.

Response to reviewer #2: Thank you dear reviewer for your helpful comments, We have now revised and restated the content as per suggestion.

9.Strength and limitations of the study: the convenience sampling technique presents a limitation towards generalising the study result. Given the selection bias, those who participated may be significantly different from those who did not.

Response to reviewer #2: Thank you, dear reviewer. We have now revised and restated the content as per suggestion.

Reviewer #3 Comments and response

Reviewer #3: This is a work about the prevalence and determinants of ED among T2D patients at Selected Government Hospitals in Gurage Zone. Anyway, no novelties about this topic are producted. I have comments and suggestions.

Abstract

- please, as suggested by international guidelines, replace here and in the manuscript, diabetic with subjects/patients with diabetes (T2D).

Response to reviewer #3: Thank you, We have now revised and replaced the abstract according to the guideline in the manuscript

Introduction

- "it remains uncertain whether erectile dysfunction (ED) in diabetic males stems

solely from hyperglycemia and microvascular complications or arises from a combination of

factors such as cardiovascular diseases, hypertension, smoking, and obesity". Vs 2010, now we have more scientific data to prove these combinations.

Response to reviewer #3: Thank you, dear reviewer. We have now included the content as per suggestion

Materials and methods ( very long! )

Response to reviewer #3: Thank you for pointing this out. We have tried to minimize the emphasized

portions.

Inclusion/exclusion criteria: You insert T2D as an exclusioon criteria. Please, check!

Response to reviewer #3: Thank you, We have now removed from exclusion criteria

Discussion

- some parts of results are here too.

Response to reviewer #3: Thank you, We have now removed parts of results

- please, clarify better strenghts and limitations of this work, not after conclusions section.

Response to reviewer #3: Thank you dear reviewer for these helpful pointers. We have now better clarified strengths and limitations of this work and inserted before conclusions section.

---

## [Decision Letter · Decision Letter 1]

18 Oct 2024

PONE-D-24-27116R1Prevalence and Determinants of Erectile Dysfunction among Type 2 Diabetes Mellitus patients at Selected Government Hospitals in Gurage Zone PLOS ONE

Dear Dr. ABDLSHIKURE ,

Thank you for submitting your manuscript to PLOS ONE. After careful consideration, we feel that it has merit but does not fully meet PLOS ONE’s publication criteria as it currently stands. Therefore, we invite you to submit a revised version of the manuscript that addresses the points raised during the review process.

Hello,

The authors need to address the important points raised by our reviewers even though they have revised many of the previous aspects.

1. In the results section,  the prevalence of ED in a previous study used for their calculation of sample size as 84.39%; they affirm that it is important to include the decimal places. However, they fail to provide such information here. The information in both Figure 2 and the text is incomplete or rather incorrect and misleading, which requires correction. For instance, the “no ED as 48 (23%)” should be “no ED as 48 (23.5%)”. The authors cannot trivialise this matter, given the status of this journal.

2. The authors did not appear to recognise that the discussion section is to interpret the study results (what do findings mean in the context of current evidence, practice and policy?) and not to state ODDS RATIO or sections of the result here. 

Please address the same before acceptance. Thanks

We look forward to receiving your revised manuscript.

Kind regards,

Kamal Sharma

Academic Editor

PLOS ONE

Journal Requirements:

Additional Editor Comments (if provided):

Hello,

The authors need to address the important points raised by our reviewers even though they have revised many of the previous aspects.

1. In the results section, the prevalence of ED in a previous study used for their calculation of sample size as 84.39%; they affirm that it is important to include the decimal places. However, they fail to provide such information here. The information in both Figure 2 and the text is incomplete or rather incorrect and misleading, which requires correction. For instance, the “no ED as 48 (23%)” should be “no ED as 48 (23.5%)”. The authors cannot trivialise this matter, given the status of this journal.

2. The authors did not appear to recognise that the discussion section is to interpret the study results (what do findings mean in the context of current evidence, practice and policy?) and not to state ODDS RATIO or sections of the result here.

Please address the same before acceptance. Thanks

Reviewers' comments:

Reviewer's Responses to Questions

**Comments to the Author**

1. If the authors have adequately addressed your comments raised in a previous round of review and you feel that this manuscript is now acceptable for publication, you may indicate that here to bypass the “Comments to the Author” section, enter your conflict of interest statement in the “Confidential to Editor” section, and submit your "Accept" recommendation.

Reviewer #1: (No Response)

Reviewer #2: (No Response)

Reviewer #3: All comments have been addressed

2. Is the manuscript technically sound, and do the data support the conclusions?

Reviewer #1: (No Response)

Reviewer #2: Partly

Reviewer #3: Yes

3. Has the statistical analysis been performed appropriately and rigorously?

Reviewer #1: (No Response)

Reviewer #2: Yes

Reviewer #3: Yes

4. Have the authors made all data underlying the findings in their manuscript fully available?

Reviewer #1: (No Response)

Reviewer #2: No

Reviewer #3: Yes

5. Is the manuscript presented in an intelligible fashion and written in standard English?

Reviewer #1: (No Response)

Reviewer #2: Yes

Reviewer #3: Yes

6. Review Comments to the Author

Reviewer #1: with respect

I am very grateful to be allowed to be a reviewer of the valuable Plos One journal.

I enjoyed reading this article. A well-chosen topic that is useful.

Here are some things to improve the article, which I hope will be corrected before acceptance:

- Is it necessary to include the word Gurage in the keyword section?

In the introduction, at the end of the second paragraph, 1.3 million diabetics should be added instead of 1.3 million diabetes.

- In the text of the article, it is better to mention diabetes in one form, for example, diabetes mellitus, or preferably after the first time as DM.

- According to the article's cross-sectional nature, the abstract's results should mention only association instead of saying that these factors were known as predisposing factors leading to erectile dysfunction.

- In the exclusion criteria, "mentally incompetent" has spelling forms that should be corrected.

- Why is type 2 diabetes written in the exclusion criteria? This is an apparent mistake that needs to be corrected because you have precisely done the study on people with type 2 diabetes. Maybe you mean type 1 diabetes?

- As it was said about the results in the article's abstract, in the discussion about the factors associated with erectile dysfunction, do not use terms that mean causality and only mention the association, for example, about age.

- Please write down which edition of ADA you used to define diabetes control. Although the definition may not have changed clearly, you should clarify which version you used.

- In the analyses, if available, it would have been better to use the continuous values of the variables to check their relationship with erectile dysfunction.

Reviewer #2: 1. In the results section, the authors have not corrected the summarised results I mentioned. For instance, the authors cited the prevalence of ED in a previous study used for their calculation of sample size as 84.39%; they affirm that it is important to include the decimal places. However, they fail to provide such information here. The information in both Figure 2 and the text is incomplete or rather incorrect and misleading, which requires correction. For instance, the “no ED as 48 (23%)” should be “no ED as 48 (23.5%)”. The authors cannot trivialise this matter, given the status of this journal.

2. The authors did not appear to recognise that the discussion section is to interpret the study results (what do findings mean in the context of current evidence, practice and policy?) and not to state ODDS RATIO or sections of the result here. This issue was not resolved in their response and should also not be trivialised.

Reviewer #3: All the queries were fully replied. Please, revise better the English language before to complete the sumbission.

7. PLOS authors have the option to publish the peer review history of their article (what does this mean? ). If published, this will include your full peer review and any attached files.

**Do you want your identity to be public for this peer review?** For information about this choice, including consent withdrawal, please see our Privacy Policy .

Reviewer #1: No

Reviewer #2: **Yes: ** Godpower Chinedu Michael

Reviewer #3: No

---

## [Author Response · Author response to Decision Letter 2]

26 Oct 2024

Academic Editor comment and response

Kamal Sharma

Academic Editor

PLOS ONE

Journal Requirements:

Response:- Thank you for the reminder regarding the reference list. We have thoroughly reviewed all the references to ensure their completeness and accuracy. We revised the reference based on the following considerations:

1. Relevance and Timeliness: The new reference provides more recent data and insights that align better with the scope of our research, ensuring that our work reflects the most up-to-date findings.

2. Improved Contextual Fit: Upon re-evaluating the original reference, we found that the newly cited work offers a clearer and more relevant theoretical or empirical basis for our arguments. This enhances the clarity and rigor of our manuscript.

3. Methodological or Regional Relevance: The new reference reflects a methodology, context, or geographic focus that aligns more closely with the objectives of our study.

We are confident that this change strengthens the manuscript and improves its overall coherence. Thank you again for your thorough review and constructive suggestions.

If the journal identifies any further issues with the references, we are happy to address them promptly.These updates are reflected in the revised manuscript and reference list.

Reviewer comments and responses

Reviewer #1:

with respect

Comment-1: I am very grateful to be allowed to be a reviewer of the valuable Plos One journal.

I enjoyed reading this article. A well-chosen topic that is useful.

Here are some things to improve the article, which I hope will be corrected before acceptance:

- Is it necessary to include the word Gurage in the keyword section?

Response-1: Thank you for your observation. We have revised the relevance of including the term "Gurage" in the keywords. we removed term "Gurage" and we added the word “Determinants”in the keywords.

Comment-2: In the introduction, at the end of the second paragraph, 1.3 million diabetics should be added instead of 1.3 million diabetes.

Response-2:Thank you for pointing out this mistake. We have corrected the phrase to "1.3 million diabetics" to ensure accuracy.

Comment-3- In the text of the article, it is better to mention diabetes in one form, for example, diabetes mellitus, or preferably after the first time as DM.

Response-3:Thank you for the valuable suggestion. We have revised the manuscript to ensure consistency by using the term "diabetes mellitus" at first mention, followed by the abbreviation "DM" throughout the remainder of the text.

Comment-4 : According to the article's cross-sectional nature, the abstract's results should mention only association instead of saying that these factors were known as predisposing factors leading to erectile dysfunction.

Response-4:Thank you for the insightful comment. We have revised the abstract to ensure that the results emphasize associations rather than implying causality, reflecting the cross-sectional nature of the study.

Comment-5: - In the exclusion criteria, "mentally incompetent" has spelling forms that should be corrected.

Response-5:We appreciate your careful review. The term "mentally incompetent" has been corrected and paraphrased to “Cognitively impaired”

Comment-6: - Why is type 2 diabetes written in the exclusion criteria? This is an apparent mistake that needs to be corrected because you have precisely done the study on people with type 2 diabetes. Maybe you mean type 1 diabetes?

Response-6:Thank you for catching this oversight. You are correct—our intention was to exclude individuals with type 1 diabetes. We have corrected the exclusion criteria accordingly.

Comment-7: - As it was said about the results in the article's abstract, in the discussion about the factors associated with erectile dysfunction, do not use terms that mean causality and only mention the association, for example, about age.

Response-7:We appreciate your observation. In line with your suggestion, we have revised the discussion section to avoid causal language and ensure that the associations are clearly stated, particularly regarding the relationship between dependent and independent variables.

Comment-8: - Please write down which edition of ADA you used to define diabetes control. Although the definition may not have changed clearly, you should clarify which version you used.

Response-8:Thank you for the suggestion. We have now specified the edition of the American Diabetes Association (ADA) guidelines used to define diabetes control in the methods section for clarity.We used “

Title: Standards of Medical Care in Diabetes – 2015: Summary of Revisions

Organization: American Diabetes Association (ADA)

Source: Diabetes Care

Publication Year: 2015

Volume: 38 (Supplement 1)

Page: S4

Comment-9: - In the analyses, if available, it would have been better to use the continuous values of the variables to check their relationship with erectile dysfunction.

Response-9: We appreciate this recommendation. Where appropriate, we have considered using continuous variables in the analysis. However, due to the data was collected in group or categorical form,continues values were not recorded ,making them inaccessible for continuous data analysis, we opted to categorize certain variables for this study. We will take this valuable input into account in future studies to enhance the robustness of our analysis.

Reviewer #2:

Comment-1. In the results section, the authors have not corrected the summarized results I mentioned. For instance, the authors cited the prevalence of ED in a previous study used for their calculation of sample size as 84.39%; they affirm that it is important to include the decimal places. However, they fail to provide such information here. The information in both Figure 2 and the text is incomplete or rather incorrect and misleading, which requires correction. For instance, the “no ED as 48 (23%)” should be “no ED as 48 (23.5%)”. The authors cannot trivialise this matter, given the status of this journal.

Response-1:Thank you for your detailed review and for pointing out these discrepancies. We have revised the results section to ensure consistency and have included decimal places where appropriate, including in the prevalence values. Additionally, we have corrected the information in both Figure 2 and the corresponding text to read "no ED as 48 (23.5%)" to accurately reflect the data.

Comment: 2. The authors did not appear to recognise that the discussion section is to interpret the study results (what do findings mean in the context of current evidence, practice and policy?) and not to state ODDS RATIO or sections of the result here. This issue was not resolved in their response and should also not be trivialised.

Response-2:We appreciate your reminder about the purpose of the discussion section. We have now revised this section to focus on the interpretation of our findings in light of existing evidence, practice, and policy. References to odds ratios and other detailed results have been moved to the results section to ensure that the discussion aligns with the journal’s expectations.

Reviewer #3:

Comment-1: All the queries were fully replied. Please, revise better the English language before to complete the submission

Response-1:Thank you for your positive feedback. We have carefully reviewed the manuscript to improve the clarity and correctness of the English language, ensuring it meets the journal’s standards.

---

## [Decision Letter · Decision Letter 2]

29 Nov 2024

PONE-D-24-27116R2Prevalence and Determinants of Erectile Dysfunction among Type 2 Diabetes Mellitus Patients at Selected Government Hospitals in Gurage Zone:A cross-sectional studyPLOS ONE

Dear Dr. ABDLSHIKURE ,

Thank you for submitting your manuscript to PLOS ONE. After careful consideration, we feel that it has merit but does not fully meet PLOS ONE’s publication criteria as it currently stands. Therefore, we invite you to submit a revised version of the manuscript that addresses the points raised during the review process.

We look forward to receiving your revised manuscript.

Kind regards,

Kamal Sharma

Academic Editor

PLOS ONE

Journal Requirements:

Additional Editor Comments:

Hello,

Though the revised manuscript is much better than the first version, the comments of the reviewer need to be addressed before final decision. Thanks

Reviewers' comments:

Reviewer's Responses to Questions

**Comments to the Author**

1. If the authors have adequately addressed your comments raised in a previous round of review and you feel that this manuscript is now acceptable for publication, you may indicate that here to bypass the “Comments to the Author” section, enter your conflict of interest statement in the “Confidential to Editor” section, and submit your "Accept" recommendation.

Reviewer #4: All comments have been addressed

Reviewer #5: (No Response)

Reviewer #6: (No Response)

2. Is the manuscript technically sound, and do the data support the conclusions?

Reviewer #4: Yes

Reviewer #5: Yes

Reviewer #6: Partly

3. Has the statistical analysis been performed appropriately and rigorously?

Reviewer #4: Yes

Reviewer #5: Yes

Reviewer #6: Yes

4. Have the authors made all data underlying the findings in their manuscript fully available?

Reviewer #4: Yes

Reviewer #5: No

Reviewer #6: Yes

5. Is the manuscript presented in an intelligible fashion and written in standard English?

Reviewer #4: Yes

Reviewer #5: Yes

Reviewer #6: No

6. Review Comments to the Author

Reviewer #4: This second version of the paper is a great improvement, the authors are to be commended.

The manuscript has been much improved and is in a nice condition now.

Reviewer #5: Seid Abrar A and colleagues presented a research article entitled ‘Prevalence and Determinants of Erectile Dysfunction among Type 2 Diabetes Mellitus Patients at Selected Government Hospitals in Gurage Zone: A cross-sectional study’. In the paper, they found the prevalence and contributing factors of ED in individuals with T2DM in the Gurage Zone.

some modifications need to be supplemented. Addressing the below concerns are mandatory to improve the quality of the paper, especially to support the statement the authors made:

1.In the “ABSTRACT” part Line 34, “overweight, obesity, duration of diabetes greater than 5 years, uncontrolled diabetes…” should be corrected definition. The Abstract section should be streamlined thoroughly.

2. in Page 6 Line 102, was the loss of follow-up considered in sample estimation?

3. in Page 7 Line 133, use the “2 or more bottles daily” to define “alcohol drinking” is improper. Please use the “milliliters per day” or “ml/d” to define.

4. There are obvious errors that affect the reliability of the results. in Page 11, the sum of “Educational Status” and “Occupational Status” were not 100%. in Page 12, the sum of “Duration of diabetes” was not 100%. Please check the full manuscript carefully.

5. in Page 16, two columns including the frequency of Erectile Dysfunction “no” and “yes” should be added between the “category” and “AOR” column.

6. Many errors in punctuation and paragraph formatting, please check the full manuscript carefully.

Reviewer #6: Thank you for giving me the opportunity to read and review this interesting article. It highlights an important area, which is often overlocked.

Overall concerns:

Throughout the article there is major negligence concerning punctuation (sometimes spaces are missing and sometimes they are superfluous), use of capital letters, bold letters i.e. line 5, 116), and paragraphs etc. This gives an overall impression of the article as inattentive.

The English language is not up to standards (i.e line 74, 90, 116, 117, 120, 121, 122).

Even though previous reviewers have pointed out the use of abbreviation T2DM, this is not consistent (i.e line 57, 92).

The statistical reporting does not meet guidelines from the journal. Table 2 seems to be divided in to two. CI are sometimes presented with .000 and sometimes with .00. Table 4 does not meet requirements.

Avoid words as only, high and low (i.e. line 335), who says that 68,7% is low?

Ethical concerns:

The sample size calculation does not include any drop out. Why? Did you not have any/expect any drop out, no missing values? These questions translate to the inclusion process and raises serious ethical concerns. Patients were “requested to give their consent”. Maybe it is the word requested that are not appropriate. Please revise. How many patients did not give their consent, or did all patients accept participation? Why are not written consent taken? I do not believe that the argument for only verbal consent holds as it stands.

Why does not the participants complete the questionnaires themselves, specially concerning a sensitive topic as erectile function? This could potentially affect the answers given, and no discussion are held on this in the limitations section.

As I read the statement in the manuscript the study has been approved by the institutional review board at a university? Is this an ethical committee, evaluating ethics?

As I read the manuscript there is contradict information concerning the action taken if patients presented with ED (line 132,198). Were they referred to the physician or was it up to the patient to raise the question with their physician?

Introduction:

The definition of ED does not have a reference. What is successful intercourse? Penetration? Throughout the article I find that an ambiguity if ED is a disease (line 61, 330), a complication of the diabetes (397) or could it be a symptom?

Method:

I would like a more extensive description of the setting as the study is set in a specific area. What is the Gurage zone? What characterizes this area? This would give the results bearing in other similar areas internationally.

When was the patients approached more specifically? When there were at the hospital, but why were they at the hospital? Did they seek acute care or had a scheduled appointment?

Inclusion/exclusion criteria’s:

Why are only heterosexual men in stable relationships included? No explanation or discussion are held on this matter. ED are not limited to this group but extend to men with other sexual orientation and relationships status. This inclusion criteria do not match the result section (Table 1), were 41% are reported as single (and separated and divorced are other options). It is unclear if these patients are in stable sexual relationships?

Figure 1:

You excluded patients with ED having endocrine and neurological causes. Is not diabetes an endocrine cause leading to neurological complications? Are those not the ones you sought for this study? Please explain further.

Why is Type 1 DM in another style than the other?

As previous reviewer has pointed out, turning variables that are continuous in to dichotomous, i.e. alcohol consumption, holds risks. Consumer of alcohol are set at two bottles or more a day. Two bottles of what? Please describe this further.

Line 184. How were the variables checked for missing values and distribution? What were the results?

Result:

Why report occupation? Is this relevant? If so, are those reported the only occupations? How can house wife be an occupation if there were only men in the study?

Line 204. “Almost half of them, 52.9%”. 52.9% is more than half.

Remove Figure 2 or remove the text describing the same information.

Line 264. Table 3. Are these results in line with the aim of this study? How did you gain this information?

I recommend removing line 294-304 and only present this information in a table.

Discussion:

I would like to have had a discussion concerning how your results in men with diabetes differs from the general population concerning determinants. Are not these similar to determinants in other populations in regard to age, overweight, alcohol consumption etc.?

You result concerning smoking is very interesting, and I would like a more extensive discussion on this topic. There is sound research on the effects on smoking, which could serve as a basis for this discussion.

Strengths and limitations:

This section needs further work raising limitations on methodological/ethical issues.

You report that bias may occur, but nothing on how this risk has been minimized. Please elaborate.

Conclusion:

The conclusion needs to be shorted. I recommend removing the first paragraph for start.

The conclusion on antihypertension - does this align with your results?

Line 426-427 is a valid point, but not belong in the conclusion, but should be elaborated on in the limitation section.

Minor:

Only one reference but refers to studies (line 53).

What is the definition of chronic medical illness (line 71)?

Line 112-115 are setting not sample size and sampling.

Why to different names on IIEF/SHIM. Choose one and stick with that.

7. PLOS authors have the option to publish the peer review history of their article (what does this mean? ). If published, this will include your full peer review and any attached files.

**Do you want your identity to be public for this peer review?** For information about this choice, including consent withdrawal, please see our Privacy Policy .

Reviewer #4: No

Reviewer #5: No

Reviewer #6: No

---

## [Author Response · Author response to Decision Letter 3]

3 Jan 2025

Reviewer #4:

Comment-1: This second version of the paper is a great improvement, the authors are to be commended.

The manuscript has been much improved and is in a nice condition now.

Response: Thank you for your positive and encouraging feedback. We have further refined the manuscript to address the additional comments raised by other reviewers.

Reviewer #5

Comment-1.In the “ABSTRACT” part Line 34, “overweight, obesity, duration of diabetes greater than 5 years, uncontrolled diabetes…” should be corrected definition. The Abstract section should be streamlined thoroughly.

Response-1: Thank you for your valuable suggestion. We have revised the "Abstract" to clarify the definitions and improve the structure. Specifically:

Overweight and obesity are now defined based on the World Health Organization (WHO) Body Mass Index (BMI) thresholds.

Duration of diabetes greater than 5 years" refers to patients with a documented history of diabetes exceeding five years.

Uncontrolled diabetes" has been clarified according to HbA1c levels (>7%).

Comment-2. in Page 6 Line 102, was the loss of follow-up considered in sample estimation?

Response-2: Thank you for raising this point. We have clarified in the manuscript that a 10% nonresponse rate was included during the sample size estimation to account for potential losses. This update is included in the Sample Size section.The calculated sample size was 204, and with the 10% addition, the final sample size became 211. Out of these, 3 individuals were excluded due to non-eligibility, and 4 did not provide consent. This has been clarified in the revised manuscript.

Comment-3. in Page 7 Line 133, use the “2 or more bottles daily” to define “alcohol drinking” is improper. Please use the “milliliters per day” or “ml/d” to define.

Response: Thank you for pointing this out. We have replaced the term "2 or more bottles daily" with a more precise definition. Alcohol consumption is now quantified in milliliters per day (ml/day):

The daily alcohol consumption of respondents was calculated using the average alcohol content (%/ml) of each drink multiplied by its volume (ml) and volumetric mass density (0.8 g/ml). Participants consuming more than 12 g of ethanol per day over the last six months were categorized as alcohol drinkers.

Comment-4. There are obvious errors that affect the reliability of the results. in Page 11, the sum of “Educational Status” and “Occupational Status” were not 100%. in Page 12, the sum of “Duration of diabetes” was not 100%. Please check the full manuscript carefully.

Response: Thank you for catching this discrepancy. We have thoroughly reviewed and corrected these clerical and editorial errors. The totals in these sections now accurately sum to 100%, and the revisions have been cross-checked for consistency

Comment-5. in Page 16, two columns including the frequency of Erectile Dysfunction “no” and “yes” should be added between the “category” and “AOR” column.

Response: We appreciate this suggestion. The requested columns have been added to Table 4, presenting the frequency of Erectile Dysfunction (yes/no) alongside categories and AOR values.

Comment-6. Many errors in punctuation and paragraph formatting, please check the full manuscript carefully.

Response: Thank you for your positive feedback. A meticulous review of punctuation, grammar, and formatting was conducted throughout the manuscript.

Reviewer #6

Comment-1: Thank you for giving me the opportunity to read and review this interesting article. It highlights an important area, which is often overlocked.

Overall concerns:

Throughout the article there is major negligence concerning punctuation (sometimes spaces are missing and sometimes they are superfluous), use of capital letters, bold letters i.e. line 5, 116), and paragraphs etc. This gives an overall impression of the article as inattentive.

Response: Thank you for your positive feedback. We have carefully reviewed the entire manuscript to address the errors in punctuation, paragraph formatting, and capitalization. Consistent formatting has been applied throughout the document. The manuscript has been proofread to ensure it meets language and style standards.

Comment-2: The English language is not up to standards (i.e line 74, 90, 116, 117, 120, 121, 122).

Response: Thank you for your positive feedback. We have carefully reviewed the manuscript to improve the clarity and correctness of the English language, ensuring it meets the journal’s standards.

Comment-3: Even though previous reviewers have pointed out the use of abbreviation T2DM, this is not consistent (i.e line 57, 92).

Response: Thank you for your valuable suggestion. The abbreviation "T2DM" is now used uniformly throughout the document

Comment-4: The statistical reporting does not meet guidelines from the journal. Table 2 seems to be divided in to two. CI are sometimes presented with .000 and sometimes with .00. Table 4 does not meet requirements.

Response: Thank you for your valuable suggestion. We have ensured that confidence intervals are consistently reported with two decimal places, and all tables have been reformatted to align with journal standards.

Comment-5: Avoid words as only, high and low (i.e. line 335), who says that 68,7% is low?

Response: We appreciate your careful review. The term " only, high and low" has been corrected and paraphrased

Comment-6: Ethical concerns:

The sample size calculation does not include any drop out. Why? Did you not have any/expect any drop out, no missing values? These questions translate to the inclusion process and raises serious ethical concerns. Patients were “requested to give their consent”. Maybe it is the word requested that are not appropriate. Please revise. How many patients did not give their consent, or did all patients accept participation? Why are not written consent taken? I do not believe that the argument for only verbal consent holds as it stands.

Response: We clarified that the sample size calculation included a 10% allowance for dropout and nonresponse.

The calculated sample size was 204, and with the 10% addition, the final sample size became 211. Out of these, 3 individuals were excluded due to non-eligibility, and 4 did not provide consent. This has been clarified in the revised manuscript.Written informed consent was obtained from all participants, as required by ethical standards. Verbal consent was an additional safeguard in certain situations, not a replacement. This clarification has been included in the methodology section.

Comment-7: Why does not the participants complete the questionnaires themselves, specially concerning a sensitive topic as erectile function? This could potentially affect the answers given, and no discussion are held on this in the limitations section.

Response: Thank you for your thoughtful observation. Interviewer-administered questionnaires were chosen due to potential literacy challenges among participants and to ensure completeness of responses. We recognize that this method could affect the honesty of answers, especially for sensitive topics such as erectile function. This limitation has been discussed in the revised manuscript, and alternative approaches are recommended for future studies.

Comment-8: As I read the statement in the manuscript the study has been approved by the institutional review board at a university? Is this an ethical committee, evaluating ethics?

Response: Thank you for pointing this out. Ethical clearance and approval to conduct the study were obtained from the Institutional Review Board of Jimma University, Institute of Health

Comment-9: As I read the manuscript there is contradict information concerning the action taken if patients presented with ED (line 132,198). Were they referred to the physician or was it up to the patient to raise the question with their physician?

Response: Thank you for highlighting this inconsistency. Upon review, we have clarified the protocol in the manuscript. "Patients identified with ED were informed of their condition and advised to consult their physician."line 132). This revision has been made to ensure consistency throughout the manuscript.

Introduction:

Comment-10 : The definition of ED does not have a reference. What is successful intercourse? Penetration? Throughout the article I find that an ambiguity if ED is a disease (line 61, 330), a complication of the diabetes (397) or could it be a symptom?

Response: Thank you for pointing this out. We have now provided a reputable reference to define Erectile Dysfunction (ED) to ensure clarity and accuracy. In the revised manuscript, we have clarified "successful intercourse" as referring explicitly to penile penetration resulting in sexual satisfaction. ED can indeed be understood variably depending on the context. This clarification ensures consistency and removes ambiguity.

Method:

Comment-11:I would like a more extensive description of the setting as the study is set in a specific area. What is the Gurage zone? What characterizes this area? This would give the results bearing in other similar areas internationally.

Response: Thank you for catching this oversight. We have expanded on the study setting, describing the Gurage Zone and the characteristics of the hospitals involved. This provides international readers with relevant context.

Comment-12 :When was the patients approached more specifically? When there were at the hospital, but why were they at the hospital? Did they seek acute care or had a scheduled appointment?

Response: Thank you for your insightful comment. In our study, patients with type 2 diabetes mellitus were approached during their visits to the hospital for routine follow-up appointments for diabetes management. This clarification has been added to the manuscript to provide more context.

Inclusion/exclusion criteria’s:

Comment-13 :Why are only heterosexual men in stable relationships included? No explanation or discussion are held on this matter. ED are not limited to this group but extend to men with other sexual orientation and relationships status. This inclusion criteria do not match the result section (Table 1), were 41% are reported as single (and separated and divorced are other options). It is unclear if these patients are in stable sexual relationships?

Response: Thank you for your thoughtful comments and observations. We acknowledge the importance of addressing the inclusion criteria and its implications on the generalizability of the study. While the study focused on heterosexual men with T2DM who reported active sexual activity during the study period, we recognize that limiting the sample to this group without detailed justification may exclude valuable insights from men with other sexual orientations.

We also acknowledge the inconsistency between the stated inclusion criteria and the demographics reported in Table 1, where 41% of participants are single, separated, or divorced. This discrepancy was due to a clerical error in the phrasing of the inclusion criteria, which incorrectly used the term "stable sexual activity" instead of "active sexual activity." We have corrected this error to align with the study's intent, ensuring that all participants were actively engaged in sexual activity during the study period, regardless of relationship status.

Figure 1:

Comment-14:You excluded patients with ED having endocrine and neurological causes. Is not diabetes an endocrine cause leading to neurological complications? Are those not the ones you sought for this study? Please explain further.

Response: Thank you for raising this important point. We agree that diabetes is an endocrine disorder that can lead to neurological complications and contributes to ED. The exclusion criteria aimed to remove participants with ED secondary to other endocrine or neurological causes unrelated to diabetes, such as hypogonadism, hyperthyroidism, or Parkinson's disease. This clarification has been added to the manuscript to ensure consistency and transparency in the study design.

Comment-15 :Why is Type 1 DM in another style than the other?

Response: We appreciate your observation regarding Type 1 DM different style. We have addressed the inconsistency ." Upon re-evaluating our data, we corrected this oversight and ensured uniform style.

Comment-16: As previous reviewer has pointed out, turning variables that are continuous in to dichotomous, i.e. alcohol consumption, holds risks. Consumer of alcohol are set at two bottles or more a day. Two bottles of what? Please describe this further.

Response: Thank you for pointing this out. We have replaced the term "2 or more bottles daily" with a more precise definition. Alcohol consumption is now quantified in milliliters per day (ml/day):

The daily alcohol consumption of respondents was calculated using the average alcohol content (%/ml) of each drink multiplied by its volume (ml) and volumetric mass density (0.8 g/ml). Participants consuming more than 12 g of ethanol per day over the last six months were categorized as alcohol drinkers.

Comment-17: Line 184. How were the variables checked for missing values and distribution? What were the results?

Response: Thank you for highlighting the need for clarity here. We have revised the section to specify the methods used for checking missing values (descriptive analysis ) and for evaluating distribution (the Shapiro-Wilk test. )The assumptions of the logistic regression model Hosmer-Lemeshow goodness of fit statistics were checked and satisfied. Multivariable logistic regression was used to identify potential confounding variables. Multicollinearity among independent variables was checked using variance inflation factors. The results were Hosmer and Lemshow goodness of fit (p-value = 0.44), Multicollinearity test (VIF) = 1.36

Result:

Comment-18: Why report occupation? Is this relevant? If so, are those reported the only occupations? How can house wife be an occupation if there were only men in the study?

Response: Thank you for pointing this out. Occupation was included in the study as it may reflect socioeconomic status, stress, and lifestyle factors relevant to erectile dysfunction. We acknowledge that the inclusion of "housewife" was an error, as the study included only men. This has been corrected in the revised manuscript, and the categorization of occupations has been clarified.

Comment-19 : Line 204. “Almost half of them, 52.9%”. 52.9% is more than half.

Response: Thank you for pointing out the inconsistency in phrasing. We have updated the text to correctly state that 52.9% represents more than half, ensuring consistency between the data and its interpretation.

Comment-19: Remove Figure 2 or remove the text describing the same information.

Response: We appreciate this recommendation. We understand your concern regarding redundancy. We have opted to retain the figure and removed the accompanying text to streamline the presentation of information.

Comment-20 : Line 264. Table 3. Are these results in line with the aim of this study? How did you gain this information?

Response: Thank you for your observation. The results presented in Table 3 align with the study’s aim by highlighting participants’ perceptions and practices related to ED management, which are critical for understanding barriers to treatment. This information was gathered through a structured questionnaire administered during the study, which included items designed to assess participants’ knowledge, attitudes, and practices regarding ED treatments.ED acceptance and treatment-seeking rate is one of the factors for Ed in diabetes patients. The study result is obtained from asking ED acceptance and treatment-seeking behavior of the patients and comparing frequency of the rate related to Ed vs non ed patients . Additional clarifications have been added to the methodology and discussion sections to ensure consistency and transparency

Comment-21: I recommend removing line 294-304 and only present this information in a table.

Response: We agree with your suggestion and have moved the detailed information from this section into a concise table for improved readability.

Discussion:

Comment-22: I would like to have had a discussion concerning how your results in men wit

---

## [Decision Letter · Decision Letter 3]

24 Jan 2025

Prevalence and Determinants of Erectile Dysfunction among Type 2 Diabetes Mellitus Patients at Selected Government Hospitals in Gurage Zone:A cross-sectional study

PONE-D-24-27116R3

Dear Dr. ABDLSHIKURE ,

We’re pleased to inform you that your manuscript has been judged scientifically suitable for publication and will be formally accepted for publication once it meets all outstanding technical requirements.

Kind regards,

Kamal Sharma

Academic Editor

PLOS ONE

Additional Editor Comments (optional):

Hello,

After 3 rigorous rounds of revision , the current version is well drafted and addresses most of the concerns raised by the reviewers in all the review activity.

The current version is good to go for editorial desk review prior to final acceptance

Thanks

Reviewers' comments:

Reviewer's Responses to Questions

**Comments to the Author**

1. If the authors have adequately addressed your comments raised in a previous round of review and you feel that this manuscript is now acceptable for publication, you may indicate that here to bypass the “Comments to the Author” section, enter your conflict of interest statement in the “Confidential to Editor” section, and submit your "Accept" recommendation.

Reviewer #5: All comments have been addressed

2. Is the manuscript technically sound, and do the data support the conclusions?

Reviewer #5: Yes

3. Has the statistical analysis been performed appropriately and rigorously?

Reviewer #5: Yes

4. Have the authors made all data underlying the findings in their manuscript fully available?

Reviewer #5: Yes

5. Is the manuscript presented in an intelligible fashion and written in standard English?

Reviewer #5: Yes

6. Review Comments to the Author

Reviewer #5: The manuscript has been improved and all criticisms satisfactorily answered. Some minor English mistakes still remain.

7. PLOS authors have the option to publish the peer review history of their article (what does this mean? ). If published, this will include your full peer review and any attached files.

**Do you want your identity to be public for this peer review?** For information about this choice, including consent withdrawal, please see our Privacy Policy .

Reviewer #5: No

---

## [Editor Report · Acceptance letter]

PONE-D-24-27116R3

PLOS ONE

Dear Dr. Abdlshikure,

I'm pleased to inform you that your manuscript has been deemed suitable for publication in PLOS ONE. Congratulations! Your manuscript is now being handed over to our production team.

Kind regards,

on behalf of

Dr. Kamal Sharma

Academic Editor

PLOS ONE